# Snakebite patterns in rural Sri Lanka and their implications for preventive measures

Asela Wijayasekara[1,2], Anjana Silva [2,3]*, Kosala Weerakoon[2], Subodha Waiddyanatha[2], Supun Wedasingha[4], Sisira Siribaddana[5], Geoffrey K. Isbister[3,6]

1 Department of Parasitology, Faculty of Medicine, Wayamba University of Sri Lanka, Kuliyapitiya, Sri Lanka, 2 Department of Parasitology, Faculty of Medicine and Allied Sciences, Rajarata University of Sri Lanka, Anuradhapura, Sri Lanka, 3 South Asian Clinical Toxicology Research Collaboration (SACTRC), Faculty of Medicine, University of Peradeniya, Sri Lanka, 4 Department of Pharmacology, Faculty of Medicine and Allied Sciences, Rajarata University of Sri Lanka, Anuradhapura, Sri Lanka, 5 Department of Medicine, Faculty of Medicine and Allied Sciences, Rajarata University of Sri Lanka, Anuradhapura, Sri Lanka, 6 Clinical Toxicology Research Group, University of Newcastle, Newcastle, Australia

* nkanjanasilva@gmail.com

## Abstract

### Background

Snakebite prevention is often neglected despite snake envenoming being a major problem in the rural tropics. We aimed to describe the epidemiology of snakebites in rural Sri Lanka to identify potential focused preventative strategies.

### Methods

The Anuradhapura Snakebite Cohort prospectively recruits snakebites admitted to the Teaching Hospital, Anuradhapura, in Sri Lanka. Epidemiological data on all snakebites from August 2013 to October 2014 and May 2017 to January 2023 were extracted.

### Results

There were 4708 snakebites, and 2202 were authenticated by specimen identification or serum analysis using enzyme-linked immunosorbent assay [*H. hypnale*, 988 (44·6%), *D. russelii*, 737 (33·3%), *B. caeruleus*, 101 (4·6%), *N. naja*, 62 (2·8%)]. Median age was 42y (IQR:29-54y), and 3027 were male (64·6%). There were 1659 (37·5%) snakebites in domestic gardens, 1153 (26·0%) on farmland, and 870 (19·6%) indoors. 3642/4620 (78·8%) were lower-limb bites, mainly the foot (3273; 70·9%). 1435/4671 (30·7%) occurred between 6 and 9 pm. Increased numbers of bites were reported from September to February. Of 988 *H. hypnale* bites, 781 (82·1%) occurred outdoors, 493 (51·8%) on the foot, and 252 (26·2%) on the hand. 704 (73·0%) *H. hypnale* bites occurred at home, and on the hand while cleaning the surrounding environment and collecting firewood. Of 721 *D. russelii* bites, 643 (89·2%) occurred

**Data availability statement:** Data is available as a supplementary file (S1 Appendix Data table) included in this manuscript.

**Funding:** The author(s) received no specific funding for this work.

**Competing interests:** The authors have declared that no competing interests exist.

on the foot, 310 (43·0%) on farmland and 286 (39·7%) outdoors. Of the 101 *B. caeruleus* bites, 75 (74·3%) occurred at night, and sixty (60·6%) while victims slept. Of 62 *N. naja* bites, 53 (85·5%) occurred during the day and 37 (59·7%) outdoors.

## Conclusion

We identified epidemiological findings that indicate possible avenues for prevention. Protective footwear would prevent outdoor bites, including 83% *D. russelii* bites and 52% *H. hypnale* bites. Long-handled tools would prevent *H. hypnale* bites to the hands, and safer sleeping environments would prevent *B. caeruleus* bites.

### Author summary

Snakebite is a significant, yet often neglected, public health issue. A higher number of snakebites is reported from rural Sri Lanka. We aimed to describe patterns of snakes in rural Sri Lanka and identify potential ways to prevent snakebites in the region. We looked at data from over 4,700 snakebite patients admitted to a major hospital in rural Sri Lanka from 2013 to identify the snakebite patterns related to highly venomous snakes in the region: Russell's viper, hump-nosed viper, Indian krait, and common cobra. Most identified snakebites were hump-nosed viper bites and Russell's viper bites. The highest number of bites occurred during the rainy season and during dusk and early night. Hump-nosed viper bites mainly occurred on feet and hands in domestic gardens while cleaning and collecting firewood. Russell's viper bites were mainly reported in farmlands and involved the feet. Most Indian krait bites occurred late at night while victims were sleeping. Wearing protective footwear could potentially prevent 83% of Russell's viper bites, half of hump-nosed pit viper bites, and 40% of common cobra bites. Maintaining safer sleeping environments, such as using mosquito nets, sleeping above ground level, would prevent 59% of Indian krait bites. Our findings suggest that snakebite burden in rural areas can be significantly reduced through the application of simple measures.

## Introduction

Snakebite is a serious health concern in the tropics [1]. Literature-based estimates suggest that over five million snakebites, 1.8 million envenomings, and 94,000 deaths occur each year globally, although these are likely underestimates [2]. Asia is severely affected by snakebite, and the World Health Organisation (WHO) aims to reduce disability and deaths from snakebite in Asia by 50% between 2021 and 2030 [3]. This can only occur by improving both the prevention and treatment of snakebite.

Sri Lanka has one of the highest incidences of snakebites, with 398 snakebites and 151 snake envenomings per 100,000 population annually [4,5]. Sri Lanka's

north-central province includes mainly rural farming communities. The region records twice the number of snakebites and three times the number of envenomings compared to national figures overall [4], making snake envenoming a major public health issue in the region [6–8]. There are several medically important snake species in the area, including Russell's viper (*Daboia russelii*), Merrem's hump-nosed pit viper (*Hypnale hypnale*), Indian krait (*Bungarus caeruleus*), and Indian cobra (*Naja naja*), which are encountered during daily activities [6,7,9,10]. Of these species, Russell's viper and Indian krait most frequently lead to severe morbidity and life-threatening effects [6,7].

Prevention is an essential and potentially straightforward method of reducing the burden of snakebite. The epidemiology of snakebite varies between regions and is influenced by local snake species, their distribution and ecology, as well as human factors, including social and occupational practices [11]. The infrastructure and lifestyle of the communities have changed gradually over the last couple of decades in Asia, including Sri Lanka. Additionally, the mechanisation of agriculture and the observed changes in weather patterns have influenced human interactions with the environment [12–14]. Further changes in vegetation, habitat loss, and changing climates influence snake populations [15]. Understanding all these dynamic factors and the interplay between them is essential for planning snakebite preventative activities [11]. Although many studies describe the epidemiological characteristics of snakebite [6,16,17], the majority of these studies aim to use this information to improve diagnosis and treatment, and few have focused on information that can be used for preventative strategies [18].

To address these gaps, we analysed data from the Anuradhapura Snakebite Cohort, a large prospective study in rural Sri Lanka. The primary objective of this descriptive study was to characterize patterns of snakebite exposure across different snake species and to identify potential preventive strategies targeting observed high-risk circumstances. Unlike previous retrospective or smaller studies, we focused on the species-specific characteristics of snakebites (location, bite site, activity, and time of day) to identify theoretical targets for prevention.

## Materials and methods

### Ethical statement

Ethical approval for the Anuradhapura Snakebite Cohort was obtained from the Ethics Review Committee of the Rajarata University of Sri Lanka (ERC/2012/036, ERC/2013/019). Informed written consent was obtained from all participants before recruitment. Consent for participants under the age of 18 was also obtained from their parents or guardians. Proxy consent was obtained from patients with impaired consciousness.

### Anuradhapura snakebite cohort

The Anuradhapura snakebite cohort prospectively records clinical, epidemiological and demographic data on confirmed snakebite cases presenting to the Teaching Hospital, in Anuradhapura, Sri Lanka. The diagnosis of snakebite is confirmed by the presence of fang marks and/or systemic or local clinical signs consistent with envenoming [19,20], The Teaching Hospital, Anuradhapura, is the largest tertiary care centre in the North Central Province of Sri Lanka (S1 Fig). As the primary referral centre for the region, the hospital receives transfers from smaller peripheral units as well as direct admissions. Consequently, the study population consists exclusively of individuals seeking formal allopathic healthcare and does not include victims who solely utilized traditional treatments or did not seek medical attention. The North Central Province is a large geographical area in the dry zone of Sri Lanka, primarily comprising agricultural communities. The province also has the highest incidence of snakebites in the country [4,5].

All snakebite patients over 16 years admitted (either directly or transferred) to the Teaching Hospital, Anuradhapura, were recruited to the Anuradhapura Snakebite Cohort. Recruitment spanned two phases: 1st August 2013–31st October 2014, and continuing from 1st May 2017 onwards. Although the database is ongoing, data for this overall analysis were retrieved up to 31st January 2023. The database prospectively records information on the demographics of snakebite patients, the circumstances of the bite, any treatment-seeking behaviours, pre-hospital care (first aid), hospital care

(investigations, clinical effects, complications, and treatment), and outcomes. Data are collected by trained medical and nursing graduates using an interviewer-administered questionnaire. Snake species are authenticated either by examination of snake specimens by a herpetologist (AS) or by venom detection in serum samples from patients using enzyme-linked immunosorbent assay (ELISA) [20–22]. Species identification was not based on victim or bystander testimony. Data were entered into a relational electronic database.

### Data retrieval and variable definitions

All snakebites in the Anuradhapura Snakebite Cohort reported up to 31st January 2023 were included in this analysis. Demographics, snakebite circumstances (location, time, activity the victim was involved in when the bite occurred, anatomical site of the bite, and any other related information), were extracted from the database.

To ensure clarity within the regional context, variables were explicitly defined and categorized. The activity the victim was performing at the time of the bite was recorded as a single primary activity, and multiple responses were not permitted. Locations were classified into distinct groups, including home gardens, farmlands [paddy fields (wet farming lands), Chena (traditional slash-and-burn shifting cultivation) and other farmlands (banana fields, coconut estates, etc.)], indoors, roadside, and other locations. For temporal analysis, the period from 18:00–21:00 was defined as 'dusk and early night', while seasonal trends were analysed based on calendar months. Anatomical bite sites were grouped into the foot, hand, leg and other sites.

### Data analysis

Continuous data with a normal distribution were analysed using parametric statistics. Continuous data that was not normal was analysed using non-parametric methods and summarised as the median and interquartile range (IQR). Ordinal and nominal data were described using counts and percentages. When indicating the percentage of data, we present the valid percentages of the variables, disregarding missing data on relevant variables. During the analysis, we focused on variables directly relevant to snakebite prevention in the regional context.

To ensure the accuracy of seasonal trend analysis and avoid bias from the hiatus in data collection, the assessment of seasonal patterns was restricted to a continuous five-year period from 1st January 2018–31st December 2022.

## Results

### Participant characteristics

During the study period, 4708 snakebite patients were recruited to the Anuradhapura Snakebite Cohort. This included 2202 species-authenticated snakebites [*H. hypnale*, 988 (44·6%), *D. russelii*, 737 (33·3%), *B. caeruleus*, 101 (4·6%), *N. naja*, 62 (2·8%), and non or mildly venomous snakes, 314 (14·2%)]. The remainder were bites due to unidentified snakes, 2495 (53·1%). The median age was 42 years (IQR: 29–54 years), and most of the victims were males (3027, 64·6%; Table 1, S1 Table).

The highest education level was available in 4466 participants, and 3699 (82·8%) were educated only up to school grade 10 or below. Agriculture-related activities, either part-time or full-time, were undertaken by 2747/4366 (62·9%; Table 1). There were 1604 (38·3%) farmers (Table 1). Patients were from all divisional secretariat areas (administrative areas) of the Anuradhapura district of Sri Lanka.

### Location and circumstances of the bite

There were 1659 (37·5%) snakebites that occurred in domestic gardens, 1153 (26·0%) on farmland and 870 (19·6%) indoors (Table 1, Fig 1, S2-S5 Tables). Most involved the lower limb in 3642/4620 (78·8%), mainly the foot for 3273 (70·9%), and the upper limb in 916/4620 (19·8%). More than half of the bites (2349/4513, 52·1%) occurred while the victim was walking. The largest proportion of snakebites was reported during the three hours from 6 pm to 9

**Table 1. Demographic characteristics, location of the bite, anatomical site of the bite, activity engaged while the bite occurred, and diurnal variation of the snakebites. (The denominators for some information varied according to the data availability and are indicated. Information regarding unidentified snakebites is shown in S1 Table).**

| | All snakebites (N = 4708) | Specimen-authenticated snakebites (n = 2202) | | | | |
|---|---|---|---|---|---|---|
| | | *Hypnale hypnale* (N = 988) | *Daboia russelii* (N = 737) | *Bungarus caeruleus* (N = 101) | *Naja naja* (N = 62) | Mild and non-venomous snakes(N = 314) |
| **Age (years)** | (n = 4672) | (n = 978) | (n = 729) | (n = 101) | (n = 62) | (n = 314) |
| Median | 42 | 45 | 42 | 38 | 45 | 37·5 |
| Interquartile range | 29-54 | 33-56 | 32-52 | 24·5-52 | 32·75-52·5 | 24-52 |
| Missing data | 36 | 10 | 8 | 0 | 0 | 0 |
| **Gender** | (N = 4695) | (N = 983) | (N = 735) | (N = 101) | (N = 62) | (N = 313) |
| Male | 3027 (64·6%) | 583 (59·3%) | 558 (75·9%) | 66 (65·4%) | 44 (71·0%) | 151 (48·2%) |
| Female | 1658 (35·4%) | 400 (40·7%) | 177 (24·1%) | 35 (34·6%) | 18 (29·0%) | 162 (51·8%) |
| Missing data | 13 | 5 | 2 | 0 | 0 | 1 |
| **Occupation** | (N = 4192) | (N = 926) | (N = 719) | (N = 91) | (N = 59) | (N = 286) |
| Farming | 1604 (38·3%) | 371 (40·1%) | 423 (58·8%) | 29 (31·8%) | 24 (40·7%) | 59 (20·6%) |
| Housewife | 562 (13·4%) | 186 (20·1%) | 83 (11·5%) | 13 (14·3%) | 10 (16·9%) | 58 (20·3%) |
| Manual worker | 304 (7·2%) | 83 (9·0%) | 42 (5·8%) | 8 (8·8%) | 4 (6·8%) | 28 (9·8%) |
| Student | 423 (10·1%) | 65 (7·0%) | 41 (5·7%) | 12 (13·2%) | 5 (8·5%) | 44 (15·4%) |
| Security forces | 260 (6·2%) | 43 (4·6%) | 49 (6·8%) | 9 (9·9%) | 1 (1·7%) | 20 (7·0%) |
| Other | 1039 (24·8%) | 178 (19·2%) | 81 (11·3%) | 20 (22·0%) | 15 (25·4%) | 77 (26·9%) |
| Missing data | 516 | 62 | 18 | 10 | 3 | 28 |
| **Location of bite** | (N = 4428) | (N = 965) | (N = 721) | (N = 96) | (N = 59) | (N = 296) |
| Domestic gardens | 1659 (37·5%) | 528 (54·7%) | 204 (28·3%) | 18 (18·7%) | 21 (35·6%) | 93 (31·4%) |
| Indoors | 870 (19·6%) | 176 (18·2%) | 41 (5·7%) | 59 (61·5%) | 17 (28·8%) | 159 (53·7%) |
| Farmlands | 1153 (26·0%) | 131 (13·6%) | 357 (49·5%) | 14 (14·6%) | 15 (25·4%) | 22 (7·4%) |
| *Paddy fields* | 805 (18·2%) | 83 (8·6%) | 269 (37·3%) | 4 (4·2%) | 11 (18·6%) | 12 (4·1%) |
| *Chena* | 311 (7·0%) | 42 (4·4%) | 73 (10·1%) | 10 (10·4%) | 3 (5·1%) | 10 (3·4%) |
| *Others* | 37 (0·8%) | 6 (0·6%) | 15 (2·1%) | ·· | 1 (1·7%) | ·· |
| Roadside | 363 (8·2%) | 62 (6·4%) | 68 (9·4%) | 1 (1·0%) | 4 (6·8%) | 4 (1·4%) |
| Other locations | 383 (8·7%) | 68 (7·1%) | 51 (7·1%) | 4 (4·2%) | 2 (3·4%) | 18 (6·1%) |
| Missing data | 280 | 23 | 16 | 5 | 3 | 18 |
| **Anatomical site of the bite** | (N = 4620) | (N = 976) | (N = 724) | (N = 94) | (N = 62) | (N = 312) |
| Foot | 3273 (70·9%) | 600 (61·5%) | 643 (88·8%) | 37 (39·3%) | 37 (59·7%) | 225 (72·1%) |
| Hand | 862 (18·7%) | 338 (34·6%) | 30 (4·2%) | 23 (24·5%) | 22 (35·5%) | 58 (18·6%) |
| Leg | 293 (6·3%) | 21 (2·2%) | 43 (5·9%) | 6 (6·4%) | 1 (1·6%) | 16 (5·1%) |
| Other sites | 192 (4·1%) | 17 (1·7%) | 8 (1·1%) | 28 (29·8%) | 2 (3·2%) | 13 (4·2%) |
| Missing data | 88 | 12 | 13 | 7 | 0 | 2 |
| **Activity while the bite occurred** | (N = 4513) | (N = 965) | (N = 724) | (N = 99) | (N = 60) | (N = 304) |
| Walking | 2349 (52·1%) | 445 (46·1%) | 402 (55·5%) | 17 (17·2%) | 25 (41·7%) | 191 (62·8%) |
| Agricultural work* | 456 (10·1%) | 62 (6·4%) | 211 (29·1%) | 1 (1·0%) | 5 (8·3%) | 9 (3·0%) |
| Cleaning | 339 (7·5%) | 142 (14·7%) | 46 (6·4%) | 3 (3·0%) | 3 (5·0%) | 22 (7·2%) |
| Sleeping | 296 (6·6%) | 15 (1·6%) | 8 (1·1%) | 60 (60·6%) | 2 (3·3%) | 11 (3·6%) |
| Collecting firewood | 114 (2·5%) | 74 (7·7%) | 5 (0·7%) | 1 (1·0%) | 1 (1·7%) | 4 (1·3%) |
| Other activity | 959 (21·2%) | 227 (23·5%) | 52 (7·2%) | 17 (17·2%) | 24 (40·0%) | 67 (22·1%) |
| Missing data | 195 | 23 | 13 | 2 | 2 | 10 |

Data are n (%) or Median (IQR). Total N values (column headers) represent the authenticated cases for each species. The 'n' values in specific sub-headers (e.g., occupation, activity) reflect the valid sample size used for calculating percentages. Discrepancies between N and n are explicitly listed as 'Missing data' for each variable. *Agricultural work includes land preparation, seeding, irrigation, manuring, weeding, and harvesting. Chena = traditional slash-and-burn cultivation.

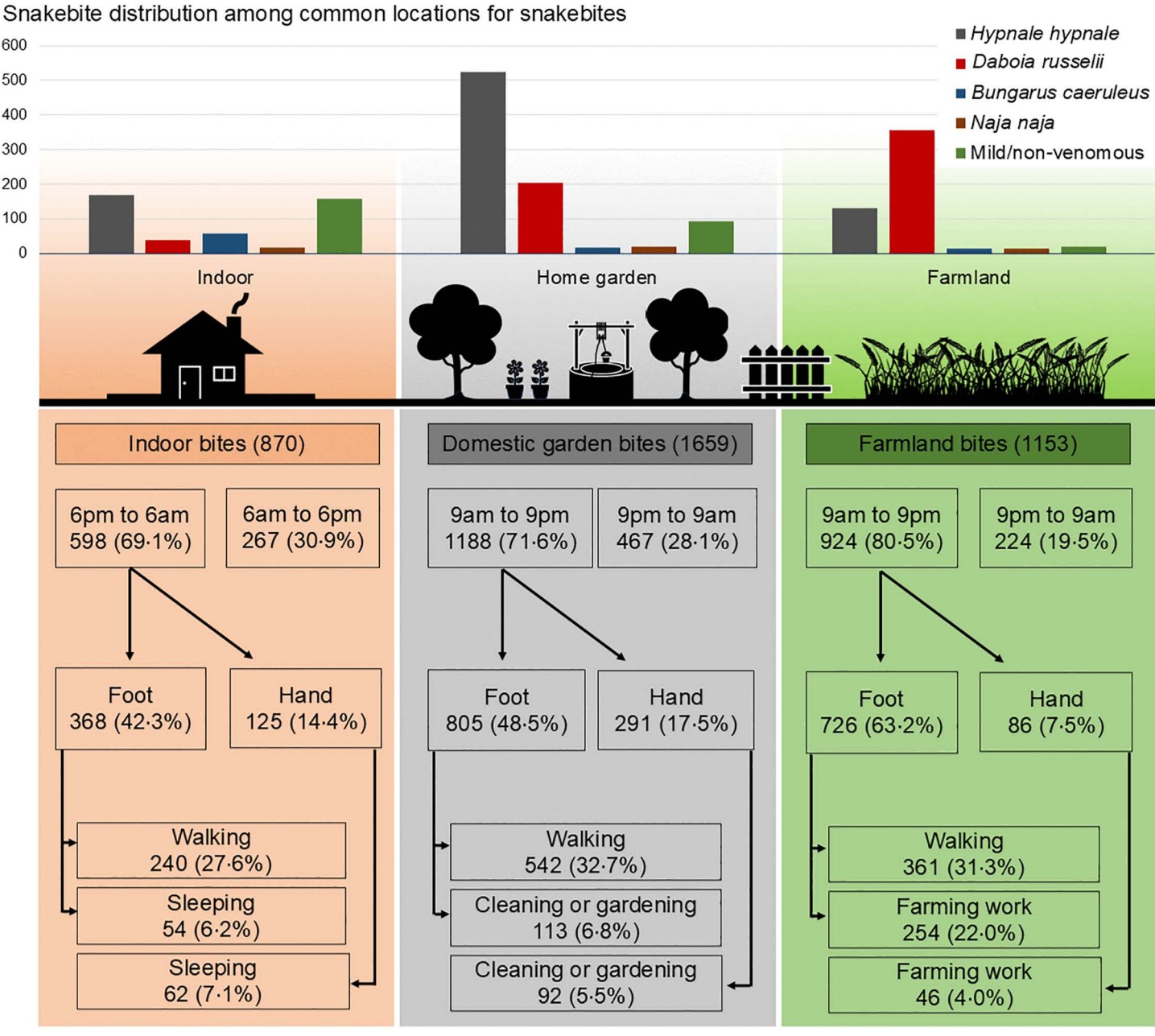

**Fig 1. Distribution and characteristics of snakebites occurring indoors, in domestic gardens, and on farmlands.** The upper panel displays the frequency and species distribution of bites across locations. The lower flow charts illustrate time, activity, and anatomical site of bites, highlighting potential targets for preventive interventions. (Percentages are calculated based on the starting N; branches may not sum to 100% as minor categories were omitted for visual clarity).

pm (1435/4671, 30·7%) (Fig 2), particularly for *H. hypnale* and *D. russelii*. Snakebites peaked from September to February (Fig 3). Snakebites that occurred on farmlands, home gardens, and indoors reached their peak during this period. Provocation of the snake was uncommon for all types of snakes (443/4708, 9·4%).

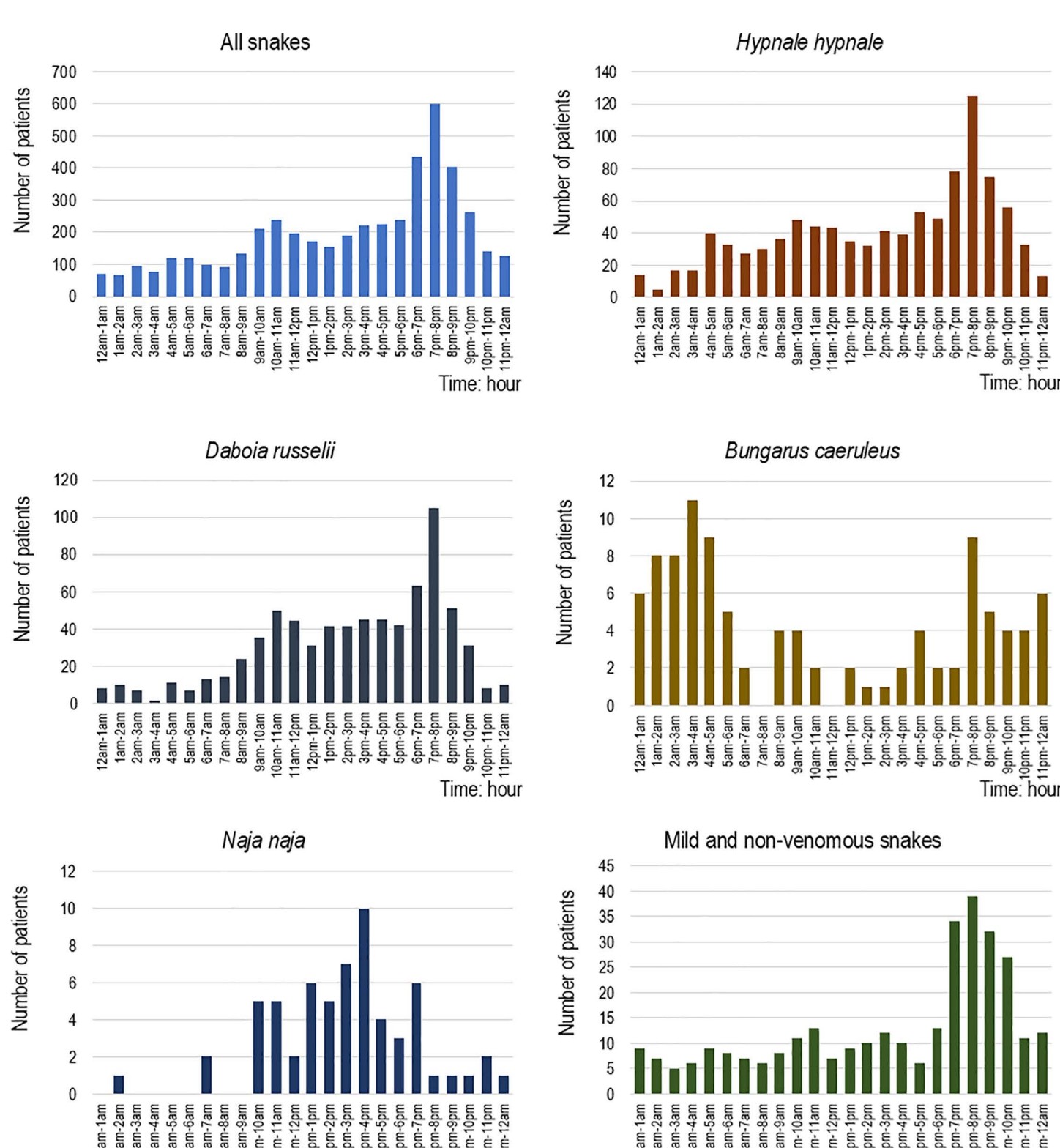

**Fig 2. Hourly variation of snakebites for all snakes and each snake species.**

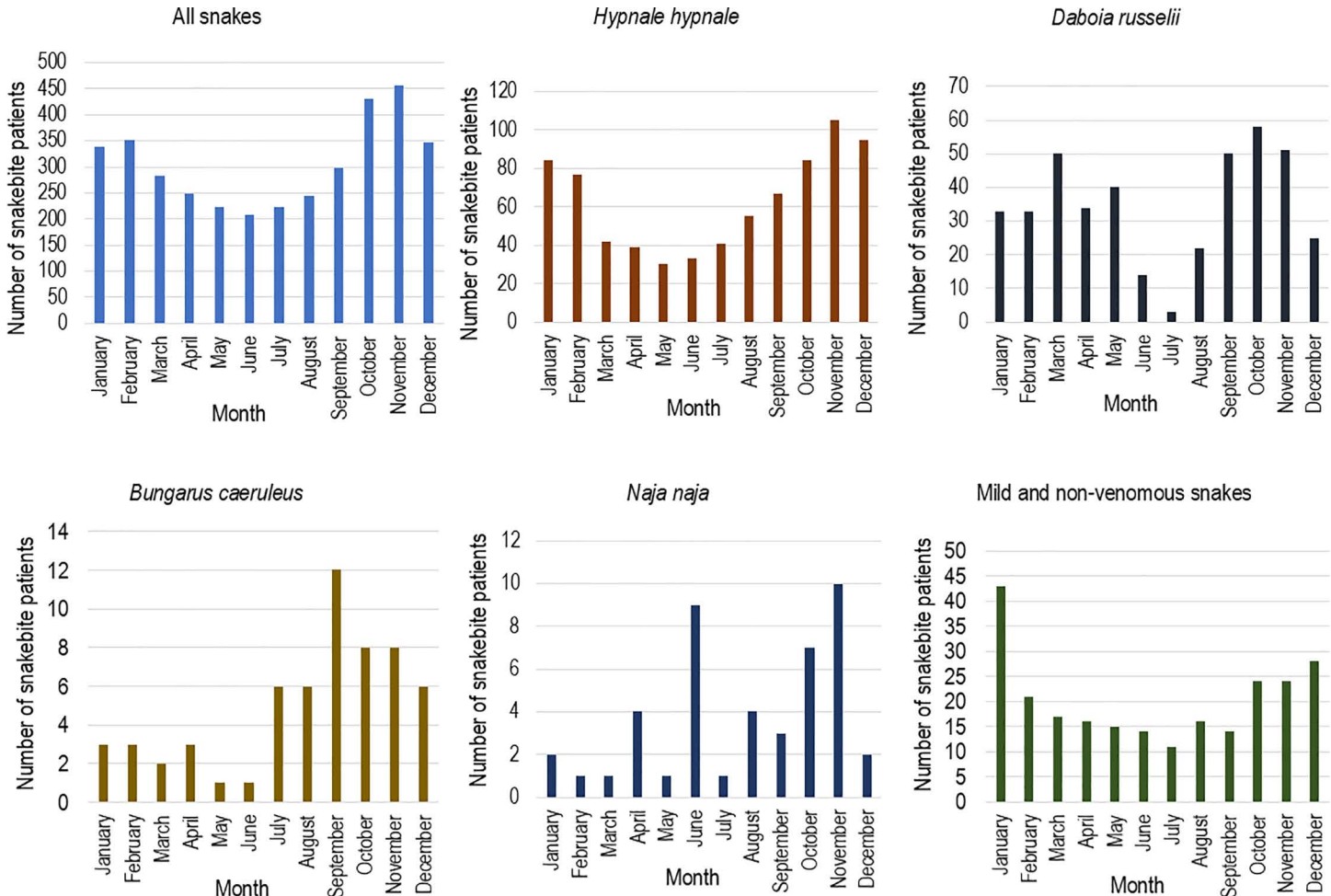

**Fig 3. Monthly variation of all snakebites and different snake species from the 1st of January 2018 to the 31st of December 2022 (the most recent five calendar years).**

### *Hypnale hypnale* (Hump-nosed viper)

Of the 988 *H. hypnale* bites, 781 (82·1%) occurred outdoors, 493 (51·8%) on the foot, and 252 (26·2%) on the hand (Fig 4). Seven hundred and four (73·0%) *H. hypnale* bites occurred in the victim's residence. Most are located on the lower limb (628/988, 63·6%) and on the upper limb (345/988, 34·9%) (Table 1). Most bites on the hand occurred while victims were cleaning (87/976, 8·9%) or collecting firewood (64/976, 6·6%). Two hundred and seventy-eight bites (28·3%) occurred from 6 pm to 9 pm (Fig 2) and *H. hypnale* bites peaked between October and February (Fig 3).

### *Daboia russelii* (Russell's viper)

Of the 721 *D. russelii* bites, 643 (89·2%) occurred on the foot, 310 (43·0%) of these on farmland and 286 (39·7%) at other outdoor locations (Fig 4). Farmlands were the most common location for bites (357/721, 49·5%) (Table 1). During agricultural work, most were exposed to *D. russelii* bites while harvesting (107/724, 14·8%), ground preparation (27/724, 3·7%)

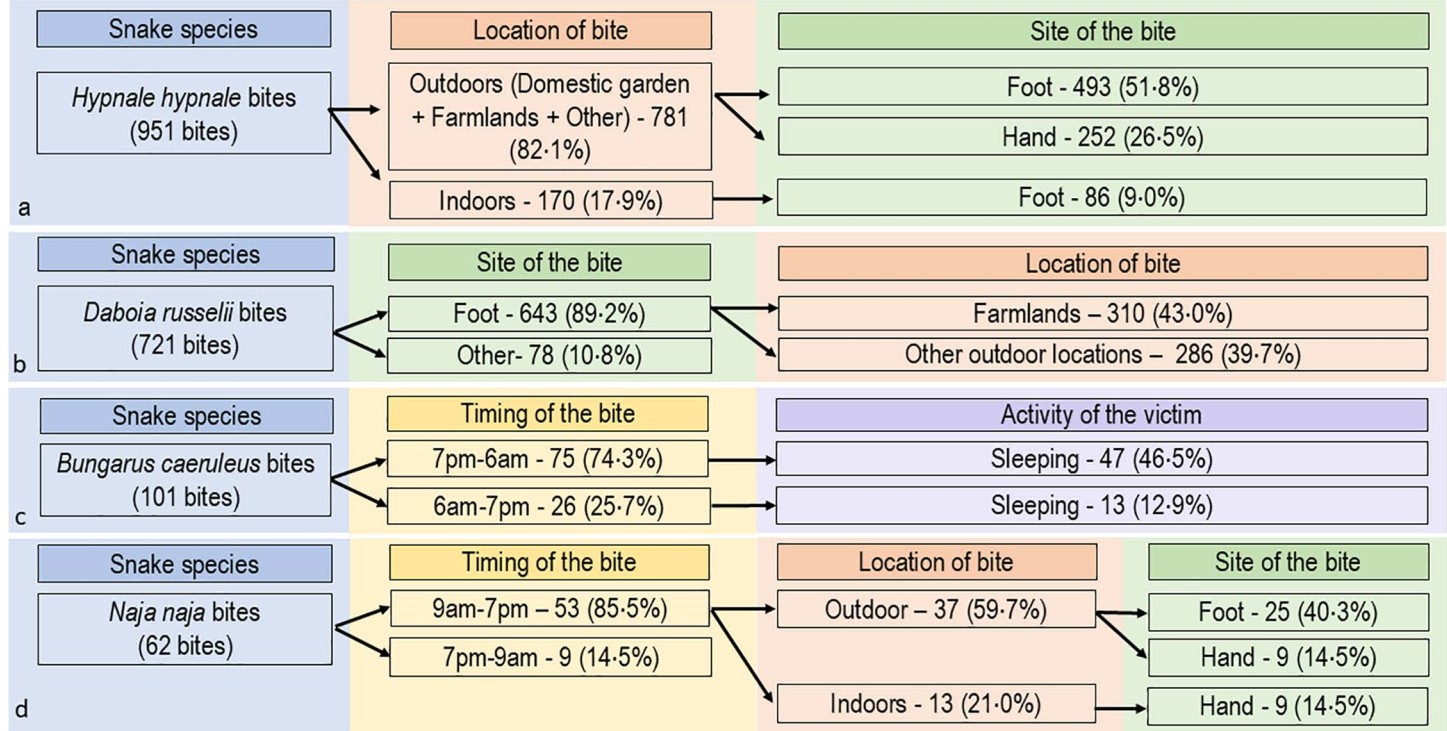

**Fig 4. Flow charts of snakebite characteristics for a,** *Hypnale hypnale,* **b,** *Daboia russelii,* **c,** *Bungarus caeruleus,* **d,** *Naja naja* **to identify potential interventions for preventing the bites.** (Percentages are calculated based on the starting N; branches may not sum to 100% as minor categories were omitted for visual clarity).

and irrigation (14/724, 1.9%). The highest proportion of *D. russelii* bites occurred from 9 am to 9 pm (593/737, 80·4%) with a peak from 6 pm to 9 pm (219, 29·7%). There were two peaks of *D. russelii* bites throughout the year, from March to May (124, 29·9%) and from September to November (159, 38·4%) (Fig 3).

### *Bungarus caeruleus* (Indian krait)

Of the 101 *B. caeruleus* bites, 75 (74·3%) occurred at night (7 pm to 6 am) (Fig 4). Sixty (60.6%) occurred while the patient slept. *B. caeruleus* peaked from 1 am to 5 am (36, 35·6%) (Fig 2). Most bites (79/101, 78·2%) were reported during the latter half of the year, peaking during September (Fig 3).

### *Naja naja* (Indian cobra)

Of 62 *N. naja* bites, 53 (85·5%) occurred during the day (9 am to 7 pm), 37 (59·7%) of these were outdoors and 25 (40·3%) on the foot (Fig 4). Most bites involved the foot (37/62, 59·7%) and occurred while walking (25/62, 41·7%). Daytime bites peaked from 12 noon to 6 pm (35/62, 56·5%).

## Discussion

We found that snakebites in Anuradhapura predominantly affect males, middle-aged individuals, and occur during outdoor activities in domestic gardens and farmlands, primarily at dusk and early night between September and February. Most

bites occurred on the foot and hand during activities such as walking, cleaning, gardening, and farming. Over three-quarters of all specimen-authenticated snakebites were due to *H. hypnale* and *D. russelii*, while bites by *B. caeruleus* and *N. naja* were uncommon.

By examining the different circumstances, we were able to establish unique circumstances and characteristics for bites by each type of snake. The majority of *H. hypnale* bites occurred outdoors, particularly in domestic gardens during the daytime, dusk and early night, when humans are most active in their domestic gardens. Bites mostly involved the feet, and the hands to a lesser extent. Farmlands were the predominant location for *D. russelii* bites (49·5%) during the daytime, dusk and early night, and almost always involved the feet. Most bites on the feet occurred while walking and during farming. Bite reports peaked during two distinct intervals: March–May and September–November, coinciding with regional agricultural seasons. Therefore, the optimal strategy is to conduct community awareness programs leading up to and throughout these high-risk periods. *B. caeruleus* bites mainly occurred at night while sleeping indoors, with most cases reported in the latter half of the year. *N. naja* bites predominantly occurred during the day in outdoor settings. Non-venomous and mildly venomous snakebites primarily occurred indoors during early nighttime hours.

Bites to the feet were particularly common with *H. hypnale* and *D. russelii*, which is because both snakes are vipers with a short body, are ambush predators, and are well camouflaged in their habitats in these locations; hence, spotting them, even during daytime, can be challenging. Our data showed that 83% of *D. russelii* bites and 52% of *H. hypnale* bites involved the feet outdoors. These could be prevented by the simple measure of wearing protective footwear, or at least covering up to the ankle level, when engaging in outdoor activities (Table 2).

**Table 2. Potential and simplest interventions for snakebite prevention and their likely impact in the region.**

| | Preventive measure | Theoretical fraction prevented* |
|---|---|---|
| 1 | Wearing protective footwear during outdoor activities to prevent bites to the feet | Up to 83% *D. russelii* bites |
| | | Up to 52% *H. hypnale* bites |
| | | Up to 40% *N. naja* bites |
| 2 | Using long-handled tools during outdoor activities to prevent hand bites | Up to 26% *H. hypnale* bites |
| | | Up to 14% *N. naja* bites |
| 3 | Maintaining a safe environment for sleeping (i.e., sleeping above ground level, applying a mosquito net) | Up to 59% *B. caeruleus* bites |
| 4 | Ensuring adequate indoor lighting to prevent snake bites to the feet | Up to 27% snakebites indoors |
| 5 | Health education before and during the high snakebite months (i.e., October to February) | Possible reduction of the large number (i.e., 53% of bites, 393 annual average snakebites in the region) of snakebites during the October to February period |
| 6 | A combination of all the above five preventive measures | Up to 88% of all snakebites |
| | | Up to 93% of *H. hypnale* bites |
| | | Up to 92% of *D. russelii* bites |
| | | Up to 95% of *B. caeruleus* bites |
| | | Up to 77% of *Naja naja* bites |
| | | Up to 81% of non and mildly venomous snakebites |

*Percentages represent the proportion of bites in the cohort occurring under circumstances modifiable by the proposed intervention.

We report larger numbers of *H. hypnale* bites and less *D. russelii* bites, compared to previous studies from the 1970s and 1980s (S6 Table) [10,23], suggesting a need for a greater focus on *H. hypnale* for prevention. This species' distribution aligns with the findings of an island-wide snakebite survey, which identified *H. hypnale* as the commonest cause of snakebites [4]. However, our cohort records a higher proportion of *D. russelii* bites (33%) than the national average, reflecting the specific ecological conditions of the dry-zone agricultural areas. Over one quarter of *H. hypnale* bites occurred on the hands while engaging in outdoor activities such as cleaning, gardening, and collecting firewood. Prevention of bites on the hand from *H. hypnale* is particularly important in reducing morbidity, because *H. hypnale* is known to cause disabling local effects in 0.7 - 18% of cases [8,24]. Using long-handled tools while cleaning and gardening, ensuring adequate light during dusk and early night, and minimising engagement in activities in the domestic garden during dusk and early night would mitigate the risk of hand bites observed during these specific activities (Table 2).

Domestic gardens, where *H. hypnale* is frequently found, play a vital role in the local community lifestyle, featuring substantial home gardening activity and the presence of water sources and toilets. Currently, Sri Lanka's snakebite prevention guidelines primarily address snakebites occurring during farming activities and do not specifically target prevention measures for home gardens and domestic settings [25]. This study emphasises the necessity to adapt these guidelines to include preventive strategies for snakebites in residential environments as well.

In our cohort, the frequency of bites for each medically important snake species increased during the northeast monsoon rainy season. Further, most snakebites were reported from domestic gardens and farmlands during these periods, indicating a temporal overlap between human agricultural activities and favourable habitat conditions for snakes in this area. Additionally, the distribution of snakebites in common locations (indoors, domestic gardens, and farmlands) indirectly suggests that snakes are abundant in the environment, and human activities in these areas make them prone to snakebites.

*D. russelii* bites were predominant in farmlands, with approximately three-quarters of the bites involving the foot (S4 Table). Considering specified agricultural activities, most victims were exposed to the snakebites during harvesting and ground preparation. This observation is notable given that these activities are predominantly mechanised within the study region [26], suggesting that exposure to snakebites persists even with modern farming practices. Therefore, wearing protective footwear is advisable even during mechanised agricultural activities (Fig 5A, B).

The number of *B. caeruleus* bites has decreased from approximately 9% of total snakebite admissions at Teaching Hospital Anuradhapura in the late 1990s [7], to only 2·2% in our study at the same hospital. This represents a notable temporal shift of the epidemiological profile in the region. While viper bites (linked to engaging in outdoor and farming activities) have persisted, krait bites (linked to housing conditions) have declined. Improving housing conditions and avoiding sleeping on the ground over the intervening two decades may have contributed to this gradual reduction in *B. caeruleus* bites (Fig 5C). There is evidence that sleeping above the ground level reduces the number of *B. caeruleus* bites in the dry zone [27]. We observed that approximately 60% of *B. caeruleus* bites occurred while the victim was sleeping, suggesting that maintaining a safer sleeping environment remains a critical priority. Considering the higher mortality rates associated with *B. caeruleus* envenoming, this will likely reduce the healthcare burden.

Similarly, *N. naja* bites have decreased from studies in the 1980s [9,10], with only 1·3% of snakebite admissions being due to *N. naja* bites in our study. We found that most bites occurred on feet and hands during the daytime at the victim's compound. The highest proportion of hand bites among authenticated snakebites was reported in *N. naja* bites. Our findings suggest that wearing protective footwear and using long-handled tools will prevent *N. naja* bites during outdoor activities (Table 2).

These findings directly support the World Health Organization's strategy for snakebite prevention and control [3], specifically the strategic pillar of 'empowering communities.' By shifting focus from generic advice to species-specific targets (e.g., footwear for vipers vs. safe sleeping for kraits), these data facilitate the development of culturally relevant,

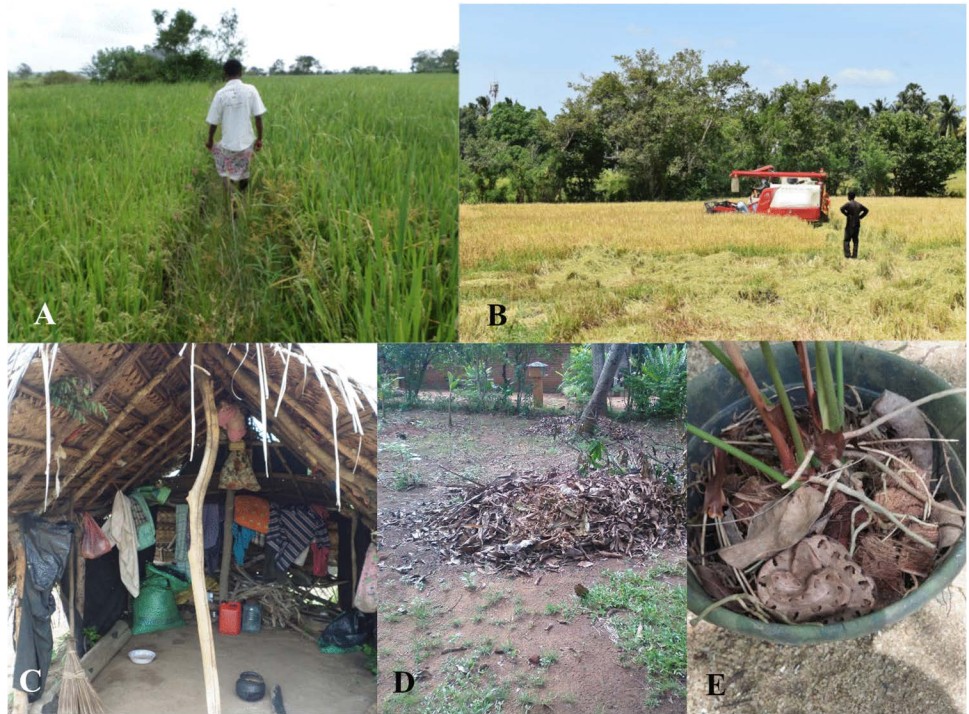

**Fig 5. Exposure to snakebite in farmlands and home gardens in rural Sri Lanka. (A)** A barefoot farmer walks in his paddy field, on which he was bitten by a *D. russelii* previously; **(B)** A mechanical harvester harvesting paddy while a barefoot farmer supervises the operation; **(C)** A hut in a farmland where a *B. caeruleus* had bitten a farmer who was sleeping on the floor; **(D)** A typical home garden with leaf litter piles providing habitat for *H. hypnale*; **(E)** *H. hypnale* that had bitten the hand of a female who was moving this plant pot in her home garden, without noticing the snake.

community-based interventions. Although this study is confined to one region in Sri Lanka, the findings are applicable anywhere with snakebite using the same principles. Many agricultural practices and domestic housing are similar across South Asia [28], and in other parts of the rural tropics, and vipers occur throughout all these regions [29]. Research conducted in similar geographical regions has demonstrated the efficacy of certain preventive practices, such as ensuring a safe sleep environment [27,30]. However, further studies are required in different regions, using a similar approach, to determine the characteristics of the snakes in each region, so that preventative measures can be developed. To translate the specific findings of this study into practice, we have outlined a framework for implementation and future community engagement (Fig 6).

This study did not capture all snakebites occurring in the district, as cases with mild or no envenoming may have been managed at secondary treatment centres without transfer. Consequently, the species distribution reported here mainly reflects the burden of medically significant snakebites requiring specialized management. While this study captures a substantial number of non-venomous or mildly venomous snakebite (n = 314), these may still be under-represented. Additionally, snakebite deaths occurring before hospital admission were not included in this study. Therefore, the patterns described here may not completely reflect the overall snakebite epidemiology in the Anuradhapura district. However, we believe the majority of significant cases were included and the epidemiological patterns described here represent the priority targets for preventing severe morbidity and mortality. When generalising the results of this study to the Sri Lankan dry zone, it is crucial to consider the regional variation in geography, climate, and human activities of local communities, as regional variation is observed even within the dry zone of Sri Lanka.

**Phase 1: Community Assessment** ONGOING

Moving beyond descriptive epidemiology, community-based cross-sectional surveys are being conducted among high-risk populations in Anuradhapura district to establish baseline data on preventive practices and behavioral barriers.
**Objective:** To quantify current adoption rates of evidence-based preventive measures and identify socio-behavioral barriers to implementation.
  *Example focus: Protective footwear utilization during agricultural activities and determinants of non-compliance despite known D. russelii exposure risk.*

*Data collection in progress (2025-2026)*

**Phase 2: Analysis of Socio-behavioural factors and feasibility for Stakeholder** PLANNED

Multi-stakeholder feasibility studies will assess implementation pathways through existing community infrastructure and government programs.
**Objective:** To map sustainable integration opportunities and understand context-specific determinants of preventive behavior adoption across agricultural and domestic settings.
**Key stakeholders:**
  * High-risk populations (farmers, rural residents)
  * Agricultural officers at ground level (for safety message integration)
  * District administrative officers (for policy and resource mobilization)
  * Educational institutions (for community outreach platforms)

*Projected timeline: 12–24 months post Phase 1*

**Phase 3: Intervention Co-Design and Pilot Testing** PLANNED

Evidence-informed interventions will be co-designed with communities and piloted to assess acceptability, feasibility, and preliminary effectiveness.
**Objective:** To develop and test culturally appropriate, context-specific prevention strategies informed by Phases 1–2 findings.
**Intervention examples:**
  * Climate-appropriate protective footwear tailored for dry zone agricultural work
  * Targeted educational campaigns aligned with identified seasonal peaks (October–January, April–May) and high-risk activities
  * Developing long-handled tools and improved lighting in domestic gardens

*Projected timeline: 24–36 months post Phase 1*

**Phase 4: Policy Translation and Integration** PLANNED

Synthesized evidence from Phases 1–3 will be translated into actionable prevention frameworks for policy integration at the district and national levels.
**Objective:** To establish an evidence-based, culturally validated snakebite prevention framework for Sri Lankan dry zone with clear implementation pathways.

*Projected timeline: 36–48 months post Phase 1*

**Expected outcome: Dissemination to the Ministry of Health, the Department of Agriculture, and provincial health authorities to incorporate findings into national snakebite prevention guidelines and agricultural safety protocols.**

**Fig 6. Proposed translational research framework for evidence-based snakebite prevention in the Sri Lankan dry zone.**

## Supporting information

**S1 Table. Comparison of bite patterns of unidentified snakes with identified snakes.**
(PDF)

**S2 Table. Snakebite patterns in common bite locations with relation to the rainy season and non-rainy season.**
(PDF)

**S3 Table. Snakebite patterns in common bite locations with relation to the daytime and nighttime.**
(PDF)

**S4 Table. Analysis based on the highest education level.**
(PDF)

**S5 Table. Anatomical site of snakebites in farmlands with type of agricultural activity.**
(PDF)

**S6 Table. Studies describing snakebite proportion of snake species in dry zone, Sri Lanka.**
(PDF)

**S7 Table. Findings of studies describing snakebite patterns of venomous snake species in Anuradhapura.**
(DOCX)

**S1 Fig. Location of the study site. Teaching Hospital Anuradhapura is located in Anuradhapura District, North Central Province, Sri Lanka (8.3248°N, 80.4140°E).** Map data: Administrative boundaries from GADM (https://gadm.org).
(TIF)

**S1 Appendix. Raw data file used for the study.**
(ZIP)

## Acknowledgments

We are grateful to the staff of the medical wards in Teaching Hospital, Anuradhapura, for their support in conducting this study.

## Author contributions

**Conceptualization:** Anjana Silva, Geoffrey K. Isbister.

**Data curation:** Asela Wijayasekara, Anjana Silva, Kosala Weerakoon, Subodha Waiddyanatha, Supun Wedasingha, Sisira Siribaddana.

**Formal analysis:** Asela Wijayasekara, Anjana Silva, Kosala Weerakoon.

**Funding acquisition:** Geoffrey K. Isbister.

**Investigation:** Asela Wijayasekara.

**Methodology:** Asela Wijayasekara, Anjana Silva, Kosala Weerakoon, Geoffrey K. Isbister.

**Project administration:** Anjana Silva.

**Resources:** Anjana Silva.

**Software:** Asela Wijayasekara.

**Supervision:** Geoffrey K. Isbister.

**Validation:** Anjana Silva, Subodha Waiddyanatha, Supun Wedasingha.

**Visualization:** Asela Wijayasekara.

**Writing – original draft:** Asela Wijayasekara.

**Writing – review & editing:** Anjana Silva, Kosala Weerakoon, Sisira Siribaddana, Geoffrey K. Isbister.

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
