## [Decision Letter · Decision Letter 0]

6 Jan 2026

Snakebite patterns in rural Sri Lanka and their implications for preventive measures.

Dear Dr. Silva,

Thank you for submitting your manuscript to PLOS Neglected Tropical Diseases. After careful consideration, we feel that it has merit but does not fully meet PLOS Neglected Tropical Diseases's publication criteria as it currently stands. Therefore, we invite you to submit a revised version of the manuscript that addresses the points raised during the review process.

Please submit your revised manuscript within by Mar 07 2026 11:59PM. If you will need more time than this to complete your revisions, please reply to this message or contact the journal office at plosntds@plos.org. Please include the following items when submitting your revised manuscript:

We look forward to receiving your revised manuscript.

Kind regards,

Wuelton Monteiro, Ph.D.

Section Editor

Wuelton Monteiro

Section Editor

Shaden Kamhawi

co-Editor-in-Chief

Paul Brindley

co-Editor-in-Chief

**Journal Requirements:**

At this stage, the following Authors/Authors require contributions: Asela Wijayasekara, Anjana Silva, Kosala Weerakoon, Subodha Waiddyanatha, Supun Wedasingha, Sisira Siribaddana, and Geoffrey Isbister. Please ensure that the full contributions of each author are acknowledged in the "Add/Edit/Remove Authors" section of our submission form.

2) Tables should not be uploaded as individual files. Please remove these files and include the Tables in your manuscript file as editable, cell-based objects. For more information about how to format tables, see our guidelines:

https://journals.plos.org/plosntds/s/tables

3) Some material included in your submission may be copyrighted. According to PLOSu2019s copyright policy, authors who use figures or other material (e.g., graphics, clipart, maps) from another author or copyright holder must demonstrate or obtain permission to publish this material under the Creative Commons Attribution 4.0 International (CC BY 4.0) License used by PLOS journals. Please closely review the details of PLOSu2019s copyright requirements here: PLOS Licenses and Copyright. If you need to request permissions from a copyright holder, you may use PLOS's Copyright Content Permission form.

Potential Copyright Issues:

- Please confirm (a) that you are the photographer of Figure 5., or (b) provide written permission from the photographer to publish the photo(s) under our CC BY 4.0 license.

- Figure 1. Please confirm whether you drew the images / clip-art within the figure panels by hand. If you did not draw the images, please provide (a) a link to the source of the images or icons and their license / terms of use; or (b) written permission from the copyright holder to publish the images or icons under our CC BY 4.0 license. Alternatively, you may replace the images with open source alternatives. See these open source resources you may use to replace images / clip-art:

4) In the online submission form, you indicated that "Data will be available upon request to the corresponding authors.". All PLOS journals now require all data underlying the findings described in their manuscript to be freely available to other researchers, either

1. In a public repository

2. Within the manuscript itself

3. Uploaded as supplementary information.

**Reviewers' Comments:**

Reviewer's Responses to Questions

**Key Review Criteria Required for Acceptance?**

**Methods**

-Are the objectives of the study clearly articulated with a clear testable hypothesis stated?

-Is the study design appropriate to address the stated objectives?

-Is the population clearly described and appropriate for the hypothesis being tested?

-Is the sample size sufficient to ensure adequate power to address the hypothesis being tested?

-Were correct statistical analysis used to support conclusions?

-Are there concerns about ethical or regulatory requirements being met?

Reviewer #1: No new analyses required

Paper adequate

Reviewer #2: 1.The manuscript lacks sufficient justification for the geographic and population coverage. It is unclear whether the cases included are representative of all rural snakebite victims or only those seeking formal healthcare. Address selection bias explicitly and discuss how it may affect estimates of incidence, severity, and snake species distribution.

2.The method of snake species identification appears to rely largely on patient reports or indirect evidence. This raises concerns about misclassification bias, especially in rural settings where snake recognition is unreliable. Clarify whether expert verification, photographs, or venom detection methods were used.

Reviewer #3: Objectives are SMART

Yes methodology is appropriate

Clearly outline the sample population

Reviewer #4: 1. Line 116. Please consider showing the study site (the hospital) within Sri Lanka using a map.

2. Line 119-125. Please add clearer definitions for key variables: time of day (including what "dusk" means), season/month groupings, activity (whether one main activity was recorded or multiple responses were allowed), location categories (e.g., garden/farmland/paddy/chena-please define "chena"), and bite site grouping (hand/foot/other).

3. Line 123-125. Species-specific analyses appear to rely on the specimen/ELISA-authenticated subset. However, approximately 53% of cases did not have species identification reported. Please explain why many cases were not authenticated (e.g., no specimen, unidentifiable specimen, no sample/ELISA) and consider a brief comparison of authenticated vs non-authenticated cases (age/sex, indoor/outdoor, time of day, bite site) to assess potential selection bias.

4. Line 118-119. Recruitment spans Aug 2013-Oct 2014 and then resumed from May 2017; please state the end date alongside these periods (you later mention Jan 2023, but it is clearer to present the full timeframe in one place). Please also note how this pause may affect interpretation of seasonal patterns and trend comparisons over time, and briefly describe how you addressed this in the analysis.

5. Line 134-136. You report "valid percentages" excluding missing data; please also report the amount of missing data and denominators (n/N) for key variables later in the result ( and tables, figures).

**Results**

-Does the analysis presented match the analysis plan?

-Are the results clearly and completely presented?

-Are the figures (Tables, Images) of sufficient quality for clarity?

Reviewer #1: Yes

Reviewer #2: yes

Reviewer #3: Yes all appropriate

Reviewer #4: 6. Table 1.

a. Sample sizes (N) differ across variables; please explain this clearly in the caption or footnotes, and report denominators consistently (n/N) with missingness highlighted.

b. Location categories appear overly granular and some (e.g., "chena") are not defined; consider collapsing categories and defining all terms in Methods.

c. Activity categories may overlap (e.g., walking during cleaning/gardening); please clarify in the Methods whether one main activity was recorded or multiple responses were allowed, and how percentages were calculated.

d. Where differences between species or categories are discussed, please either support them with simple statistical comparisons (e.g., Fisher's exact or chi-square tests) or rephrase as descriptive patterns (e.g., "in this cohort, XX% occurred" / "most frequently reported" / "the majority occurred").

7. Table 2 (in discussion).

a. It currently implies intervention effectiveness or "% preventable" (e.g., "60% potentially prevented"). As this is a descriptive study, the manuscript should avoid predicting prevention impact (e.g., "X% preventable") and instead describe prevention opportunities suggested by the observed patterns. I suggest reframing this table as "prevention opportunities/targets suggested by observed bite circumstances" and removing quantitative "% prevented" claims.

8. Figures 1 and 2. They are difficult to interpret because denominators appear to change and branches may not sum to 100%. The figures are overly complex, with multiple layered categories, which makes them difficult to interpret. Consider simplifying to stacked bar charts/heatmaps (e.g., species by location/activity/time).

9. Figure 3. The recruitment pause should be considered in this analysis. Consider presenting counts by year and month, or otherwise clarify how months/years with incomplete coverage were handled (with explanation).

**Conclusions**

-Are the conclusions supported by the data presented?

-Are the limitations of analysis clearly described?

-Do the authors discuss how these data can be helpful to advance our understanding of the topic under study?

-Is public health relevance addressed?

Reviewer #1: Yes - but i would like more information on obtaining impact. otherwise risks just being a paper that describes and does not change anything

Reviewer #2: yes

Reviewer #3: Limitations not clearly stated especially the number of deaths as a result of snakebites

Reviewer #4: 10. Discussion (overall). The Discussion contains several statements that read as causal or risk-factor conclusions without supporting analysis. Please soften wording to reflect descriptive patterns and clearly state that prevention implications are potential targets rather than quantified effects.

11. Limitations (Line 314-320). On the current limitation statement, please also add some issues such as: hospital-based selection, species authentication in only a subset, interrupted recruitment affecting seasonality, and potential recall or category misclassification for activity, time, and location.

12. The aim (Line 101-105) is appropriate for the sample size, but to match this aim, the manuscript should clearly present findings as descriptive patterns unless formal statistical comparisons are added.

**Editorial and Data Presentation Modifications?**

Reviewer #1: See below

Reviewer #2: (No Response)

Reviewer #3: minor revision

Reviewer #4: (No Response)

**Summary and General Comments**

Reviewer #1: This manuscript reports 4708 snake bites over around 7 years, of which 2202 were species authenticated and analysed more carefully. The authors identify species specific characteristics that allow them to propose interventions to reduce snake bite.

I have a few comments, mostly minor

Major

1. It is easy to make recommendations, much more difficult to implement things. The authors have identified several interventions, none of which are particularly surprising or novel. I would like to know what overall number (%) of bites would be prevented by implementing their 5 recommendations

More importantly I would like to know how they intend to implement their recommendations? With whom, over how long? Does it needs funding. I would really like to know that they have this in process already; otherwise it risks becoming another epi paper that does not result in change or patient benefit. A Text box or figure in the Discussion laying out plans and time-lines would markedly increase the importance of the paper, especially when it is done and the effects are measured

2. the abstract needs mildly improving (with implications for main text as well please)

a) “species-authenticated” needs defining here

b) because of the high mortality with B caeruleus, pls provide the number of cases that occurred when sleeping (to accompany the ‘walking’ data – low numbers but will save lives)

c) “and bites peaked from September to February” – can you indicate why here (was different for R viper I think)

d) “and hand bites occurred while cleaning” – cleaning what? Themselves (washing?) or brushing ground or picking up rubbish? Wld be good to get a better understanding of what action is meant by ‘cleaning’

e) what total number of bites would be prevented by the 5 interventions

f) the discussion says that thing shave changed since 1980s/90s. Would be nice to include this in the conclusions as it is important to understand change in epidemiology (authors argue reduced B caeruleus is due to change in sleeping habits)

g) conclusion starts with: “We found simple measures could prevent many snakebites.” Not really - study identified epidemiology that indicates possible avenues for prevention. This study did not actually ‘find’ simple measures to prevent outdoor bites

h) collecting firewood seems to be dangerous activity. However nmot sure how long -handed tools could help prevent these bites

3.

“The remainder were bites due to unidentified snakes, 2495 (53·1%).”

How many of these patients were envenomed by main 4 but did not have either snake brought in or positive ELISA? Pls give numbers for this group by species and by non-envenomed or mildly-envenomed

4.

Figure 1 indicates that 116 were bitten while sleeping in the house. This differs from 96 autheticated bites and the 296 in all species in table 1. Please clarify

Minor

1.

“75 (74·3%) occurred at night (7 pm-6 am), and 47 (46·5%) while slept.”

60% bites occurred while sleeping, it is just that 47% occurred during night time hours. The tight link between ‘night’ and ‘slept’ is not clear. Could separate and show that 60% occurred while sleeping, which is the main point. Were these extra bites during mid-day, when I did not think that krait were about?

Of note the author summary makes this clear “Most Indian krait bites occurred late at night while victims were sleeping.”

2.

“with 398 snakebites and 151 snake 82 envenomings per 100,000 population annually”

Wld be good to define bite vs envenoming, especially for mild snake bite. Were these patients envenomed or just bitten with local effects like pain?

3.

“The diagnosis of snakebite is confirmed by the presence of fang marks and/or systemic or local clinical signs CONSISTENT WITH ENVENOMING”

4.

The term dusk is used in the paper, sometimes to indicate 18.00 to 21.00. Dusk is around 18.00 to 19.00 in SL. 19.00 to 21.00 is dark, after nightfall. So not really dusk – people should be using light sources at that time, so there is some nuance here

5.

“More bites were reported from March to May and from September to November, which is consistent with regional agricultural seasons,”

And

“In our cohort, bites by each of the snakes occurred during the peak of the Northeast monsoon rainy season. Further,”

These seem to clash and perhaps differ from the abstract (see comment #2 above)

6.

“Prevention of bites on the hand from H. hypnale is particularly important in reducing morbidity, because H. hypnale is known to cause disabling local effects [8,24].”

Please say how common - how big a problem is it? If it is only 1 in 100, hard to make argument. If 30%+ of hand bites cause long term effects, then clearly major. Worth just adding % here to support your argument it is important

7.

Table 2 – please add in an ‘all 5 measures’ and show how much be prevented if all done

8.

Very nice illustrative photos in Figure 5

Reviewer #2: This manuscript addresses snakebite epidemiology in rural Sri Lanka, a topic of high relevance to neglected tropical diseases, occupational health, and rural public health. Given Sri Lanka’s well-documented burden of snakebite envenoming, the study has the potential to contribute meaningful evidence to guide prevention strategies and policy interventions. The dataset appears substantial, and the authors attempt to link epidemiological patterns with practical preventive implications, which aligns well with the journal’s translational focus.

However, while the study is valuable, several conceptual, methodological, and interpretative limitations reduce its current impact. In particular, the definition of novelty, clarity of study design, analytical depth, and strength of inference regarding preventive measures require significant improvement. The manuscript would benefit from a more rigorous epidemiological framework and a clearer separation between descriptive findings and policy recommendations.

1.The manuscript does not clearly define whether the study is prospective, retrospective, or cross-sectional. The authors should explicitly clarify:

What hypotheses were tested (if any)?

Whether the primary goal was descriptive epidemiology or evaluation of risk factors.

Reframe the Introduction to clearly articulate the study design, primary outcomes, and how these differ from prior snakebite studies in Sri Lanka.

2.The manuscript lacks sufficient justification for the geographic and population coverage. It is unclear whether the cases included are representative of all rural snakebite victims or only those seeking formal healthcare. Address selection bias explicitly and discuss how it may affect estimates of incidence, severity, and snake species distribution.

3.The method of snake species identification appears to rely largely on patient reports or indirect evidence. This raises concerns about misclassification bias, especially in rural settings where snake recognition is unreliable. Clarify whether expert verification, photographs, or venom detection methods were used.

4.Several key Sri Lankan and South Asian snakebite studies are cited, but comparison is largely superficial. Differences and similarities with prior national surveillance data are not sufficiently explored. Expand the Discussion to compare:

Incidence patterns

Species distribution

Temporal trends

with previous Sri Lankan studies and WHO snakebite frameworks.

5.The manuscript would benefit from professional English editing. Several sentences are long, repetitive, or unclear.

Reviewer #3: none

Reviewer #4: This manuscript uses a large hospital-based cohort to describe snakebite circumstances and, where available, specimen/venom-based identification of the biting species. The descriptive dataset has clear potential value for informing prevention messages in the region. However, the current Methods and presentation lack key variable definitions and denominators, and parts of the Results/Discussion move beyond descriptive reporting to imply associations and prevention impact without supporting analyses. Clarifying methods, improving presentation, and tempering over-interpretation would substantially strengthen the manuscript.

PLOS authors have the option to publish the peer review history of their article (what does this mean? ). If published, this will include your full peer review and any attached files.

**Do you want your identity to be public for this peer review?** For information about this choice, including consent withdrawal, please see our Privacy Policy .

Reviewer #1: No

Reviewer #2: **Yes:** Ahmed Hamdy Ghonaim

Reviewer #3: **Yes:** Melanie Abongo

Reviewer #4: **Yes:** Bolor Bold

**Figure resubmission:**
---

## [Editor Report · Decision Letter 1]

26 Feb 2026

Dear Prof. Silva,

We are pleased to inform you that your manuscript 'Snakebite patterns in rural Sri Lanka and their implications for preventive measures.' has been provisionally accepted for publication in PLOS Neglected Tropical Diseases.

Best regards,

Wuelton Monteiro, Ph.D.

Section Editor

Wuelton Monteiro

Section Editor

Shaden Kamhawi

co-Editor-in-Chief

Paul Brindley

co-Editor-in-Chief

---

## [Editor Report · Acceptance letter]

Dear Prof. Silva,

We are delighted to inform you that your manuscript, "Snakebite patterns in rural Sri Lanka and their implications for preventive measures.," has been formally accepted for publication in PLOS Neglected Tropical Diseases.

Best regards,

Shaden Kamhawi

co-Editor-in-Chief

Paul Brindley

co-Editor-in-Chief
